# Eco-Friendly Rice Straw Paper Coated with Longan (*Dimocarpus longan*) Peel Extract as Bio-Based and Antibacterial Packaging

**DOI:** 10.3390/polym13183096

**Published:** 2021-09-14

**Authors:** Rungsima Chollakup, Wuttinant Kongtud, Udomlak Sukatta, Maneenuch Premchookiat, Kanyanut Piriyasatits, Hataitip Nimitkeatkai, Amnat Jarerat

**Affiliations:** 1Kasetsart Agricultural and Agro-Industrial Product Improvement Institute (KAPI), Kasetsart University, Chatuchak, Bangkok 10900, Thailand; aaprmc@ku.ac.th (R.C.); aapwnk@ku.ac.th (W.K.); aapuls@ku.ac.th (U.S.); 2Food Technology Program, Kanchanaburi Campus, Mahidol University, Saiyok, Kanchanaburi 71150, Thailand; maneenuch.pre@gmail.com (M.P.); kanyanut.pir@gmail.com (K.P.); 3School of Agriculture and Natural Resources, University of Phayao, Muang, Phayao 56000, Thailand; hataitip.ni@up.ac.th

**Keywords:** antibacterial packaging, bio-based packaging, longan peel extracts, rice straw paper

## Abstract

This study aimed to develop active paper from rice straw fibers with its function as antibacterial activity obtained from longan (*Dimocarpus longan*) peels. The morphology and mechanical properties of fibers of rice straw were examined as quality parameters for paper production. Rice straw paper (RSP) with basis weight ca 106.42 g/m^2^, 0.34 mm thickness, 34.15% brightness, and 32.26 N·m/g tensile index was successfully prepared from fibers and pulps without chemical bleaching process. Bioactive compounds of longan peels were extracted using maceration technique with a mixture of ethanol-water, and subsequently coated onto RSP at concentration of 10%, 15% and 20% (*w*/*v*). Fourier transform infrared (FTIR) spectroscopic analysis demonstrated the functional groups of phytochemicals in the peel extract. The results of physical properties showed that the coated RSP had similar thickness and tensile index, but had lower brightness compared to control papers. Scanning electron microscopy (SEM) confirmed the significantly different of surface and cross-section structures between coated and uncoated RSP. The coated RSP had relatively greater barrier properties to prevent water absorption. In addition, the RSP coated with longan peel extracts showed significant antibacterial activity against foodborne bacteria, *Staphylococcus aureus* and *Bacillus cereus*. This study reveals the benefits of natural byproducts as potential materials for active packaging prepared by environmentally friendly processes.

## 1. Introduction

Natural constituents retrieved from agricultural wastes and byproducts can be alternative sources for bio-based and functional materials. For instance, longan (*Dimocarpus longan*) is a subtropical fruit grown in East and South East Asia, and its peels are regarded as byproducts from agro-industrial processing or consumption of the fruit itself. Annual worldwide production of longan is 3.4 million tons [1], 10% and 15% weight are accounted as peel and seed, respectively [2]. During the commercial processing of dried and canned products, longan peels and seeds are generally discarded. These fruit peels are often a waste product and effect to the environment as they are gradually fermented and release off odors. The utilization of longan peels will promote a reduction in organic waste. Furthermore, longan peels are known for their source of high value components as well as functional ingredients that benefit human health. Previous phytochemical studies have reported polyphenolic compounds of longan pericarps display ellagitannin, ellagic acid, gallic acid [3], gallic acid derivatives [4] and its conjugates, (−)-epicatechin, 4-O-methyl gallic acid, flavonoid glycoside, quercetin, and kaempferol glycoside as bioactive compounds [5]. These phenolic compounds are known for their antioxidant, antimicrobial and anti-inflammatory activities [6]. The potential application and benefits of antimicrobial agents found in different botanical extracts have been highlighted in the recent reviews [6,7]. The main sources of phenolic compounds are fruits, vegetables and woody vascular plants [5,6,7]. Natural phenolic compounds play an important role as antibacterial agents to inhibit the growth of bacteria. To overcome the potential risk of synthetic compounds, plant extracts have been extensively studied as alternative bioactive compounds having antimicrobial activity [6,7].

Rice straw rich in cellulosic fibers is obtained after harvesting of paddy rice. It has been reported that rice straw consisted of in range of 28–41% cellulose [8]. The annual global production of paddy rice is approximately 740 Mt [9]. It is estimated that for 1 kg of paddy production, about 0.41–3.96 kg of rice straw residue is generated as organic byproduct [8]. Cellulosic biomass is attractive as a natural raw material for the production of environmentally friendly paper packaging due to its renewable and biodegradable properties. In addition to its general use as secondary and tertiary packaging, paper is also used as primary packaging and placed in direct contact with food products.

Packaging production, especially the concept of blending natural bioactive compounds into packaging materials, which classified as active packaging, has been developed and focused on recently [10,11,12,13]. Edible film of whey protein isolate nanofibers emulsified with carvacrol as an antimicrobial agent and incorporating glycerol as a plasticizer was reported to have a strong inhibitory effect on the growth of *E. coli*, *Listeria monocytogenes*, *Salmonella typhimurium*, and *Staphylococcus aureus* [13]. The utilization of environmentally friendly active packaging and natural compounds could be alternative options to overcome health concerns and spread environmental awareness [12]. Recently, there has been an increasing trend of using paper as packaging, resulting in the development of paper produced from natural fibers. The effective utilization of byproducts, by translating waste and byproducts into beneficial resources through technological approaches, has been reported to increase the sustainable development of economic value and reduce environmental pollution [14]. Concerns about fruit processing, which discharges large amounts of fruit peel waste, and the agrofiber byproducts of paddy rice have also motivated us to focus our attention on this subject. It is necessary to develop efficient, eco-friendly and active packaging.

Although applications of rice straw cellulose and paper have been reported during recent years [15,16] there are few studies on the antibacterial activity of rice straw paper [17]. Therefore, we report herein the development of biobased and bioactive material by coating longan peel extract onto rice straw paper. The morphology of rice straw fiber was measured in order to confirm the quality of paper. Bioactive compounds were extracted from dried longan peels using a mixture of ethanol and water and subsequently coated onto rice straw paper to produce a paper-based functional food packaging. The longan extract coated papers were characterized using analytical methods such as FTIR (Fourier Transform Infrared) spectroscopy and scanning electron microscopy (SEM). The functional properties of coated paper, including tensile index, water absorption and antibacterial activity, were evaluated.

## 2. Materials and Methods

### 2.1. Materials

Rice straw was obtained as native biomass from extension farm of Kasetsart University, Bangkok, Thailand. Longan (*Dimocarpus longan*) peels of an organic farm was provided from Fruitdee Co., Ltd. (Bangkok, Thailand). The fruit peels were cut into small piece (≤5 mm) and dried at 45 °C for 48 h in a hot air dryer (Binder FD 115, Tuttingen, Germany). Oxidized starch, EXELSIZE 8 was obtained from SMS Corporation, Pathum Thani, Thailand. All analytical grade chemicals were purchased from Sigma-Aldrich S.A. (Madrid, Spain) and were used without further purification. Food borne pathogenic bacteria, *Staphylococcus aureus* ATCC 6538, *Bacillus cereus* ATCC 11778 and *Escherichia coli* ATCC 25922 were obtained from American Type Culture Collection (ATCC, Virginia, USA) and used as test bacteria. The bacterial strains were cultured in tryptic soy medium (Difco, Thermo Fisher Scientific Inc., Waltham, MA, USA) and stored at 4 °C for further study.

### 2.2. Extraction of Longan Peels

Dried longan peels were extracted by maceration method [18] with slight modifications. Sample was macerated in 75% (*v*/*v*) ethanol (1:10) for 72 h then the sample residues were macerated again for 3 times to maximize the extraction. The obtained supernatants were combined and concentrated under vacuum using a rotary evaporator (Buchi, Flawil, Switzerland) at 40 °C. The concentrated longan peel extract was subsequently freeze dried in a freeze dryer (Labconco, Kansas City, MO, USA) and kept at 4 °C in a dark condition in airtight containers until used.

### 2.3. Total Phenolic Content Determination

Total phenolic content was analyzed according to the method using a previous published method with slight modification [5]. One gram of longan peel extract was homogenized in 80% ethanol and centrifuged at 12,000× *g* for 20 min at 4 °C. The supernatant was mixed with 1 mL of Folin–Ciocalteu phenol solution. Then, the mixture was left standing for 5 min, after which, 3 mL of 20% (*w*/*v*) sodium carbonate was added to the mixture and incubated in the dark at room temperature for 60 min. Total phenolic content was determined spectrophotometrically at 765 nm and evaluated using a standard curve provided gallic acid. The contents were expressed as μg of gallic acid equivalent/mg of longan peel extract (μg GAE/mg).

### 2.4. High Performance Liquid Chromatography (HPLC) Analysis of Longan Peel Extract

The HPLC analysis was performed using an HP1100 HPLC equipped with a diode-array detector (Hewlett-Packard, Palo Alto, CA, USA) and a reversed phase column, LiChrospher RP-18. The compounds were eluted with a gradient system of 0.4% formic acid (solvent A): methanol (solvent B) at a flow rate of 1.0 mL/min. The column temperature was set at 25 °C. The UV detection was carried out at 270 nm. The injection volume was 10 µL. The gradient system started from 0 min (100%A) to 2 min (95%A), 5 min (70%A), 8 min (66%A), 11 min (45%A), 14 min (45%A), 17 min (100%A) and maintained at this ratio until 20 min.

### 2.5. Preparation of Rice Straw Paper (RSP)

Rice straw was cooked with 20% (*w*/*v*) NaOH solution at 100 ± 10 °C for 3 h. The ratio of rice straw to alkaline solution was at 1:10. The obtained pulps were thoroughly washed with distilled water in order to remove the remain alkaline until the pH was neutral. During washing, a disintegrator was used for disintegration of pulp suspensions. For the rice straw paper (RSP) production, 650 g of pulps were suspended and spread into 72 × 84 cm-size frame. The frame containing pulps was then air-dried at 30 °C for 1 day.

### 2.6. Preparation of Coated Longan Peel Extract Papers (Coated RSP)

For surface sizing of the RSPs, oxidized starch of 8% (*w*/*v*) was gelatinized in distilled water at 90 ± 5 °C, cooled down to 40 °C and mixed with the longan peel extract at 10, 15 and 20% (*w*/*v*). The coated RSPs were prepared by applying 4 g mixture of longan peel extract and oxidized starch onto the surface of papers (12.5 × 12.5 cm) using a steel roll rod coater, RDS 10 (Webster, NY, USA). For the starch coated RSP, the procedure was prepared as mentioned above but without an addition of the longan peel extract. Prior to analysis, the papers were conditioned at a temperature of 25 ± 2 °C with a relative humidity (RH) of 50 ± 5% for 48 h.

### 2.7. Characterization of Rice Straw Fiber and Paper

#### 2.7.1. Morphology of Rice Straw Fiber

Rice straw fibers observed under microscopy (Leica, Model LM750, Wetzlar, Germany) were measured using Leica 4.8 program. The following ancillary properties were calculated.
Felting coefficient = Fiber length/Fiber width(1)
Flexibility coefficient = Lumen width/Fiber width(2)
Runkel ratio = [2 × Thickness of cell wall]/Lumen width(3)

#### 2.7.2. Fourier Transform Infrared (FTIR) Spectroscopy

The FTIR analysis of longan peel extract and the chemical functional groups on the surface of paper samples was performed using an FTIR spectrometer (Thermo Scientific Nicolet IR200, Waltham, MA, USA). The paper samples (approximately area of 1 cm^2^) were placed in a standard sample holder. A total of 128 scans were accumulated in attenuated total reflection (ATR) mode with a resolution of 4 cm^−1^. The spectra were obtained in the range of 4000 to 400 cm^−1^.

#### 2.7.3. Physical Properties of RSP

Th percentage of brightness was measured following the standard method of TAPPI T 452 om-18 [19]. Tensile index was measured using a Schopper Tensile Tester (Kumagai Riki Kogyo Co. Ltd., Tokyo, Japan) according to TAPPI T 494 om-06 [20]. RSP samples were cut into rectangular strips (15 mm × 150 mm) and then clamped both ends of each specimen strips with the paper-based holder. The tensile testing was performed at a strain rate of 25 ± 5 mm/min and a clamp distance of 100 mm. The breaking force value (N) was recorded and used to determine the tensile index (N·m/g) according to the following Equation (4):Tensile index (N·m/g) = (653.8 × breaking force)/basis weight(4)

The thickness of the prepared papers was measured using a digital micrometer series S00014 (Mitutoyo Corporation, Kawasaki, Japan), having ±0.001 mm accuracy. Measurements were performed at five random positions and values were averaged.

#### 2.7.4. Morphology by Scanning Electron Microscopy (SEM)

The morphology of the fibers was characterized with 5 kV using scanning electron microscopy (SEM) (Hitachi SU8020, Tokyo, Japan) at 400× magnification. Fibers were mounted onto a bronze stab with carbon double-sided tape and coated with gold in a sputter coater (E1030 Ion Sputter, Hitachi, Tokyo, Japan). The RSP samples were immersed in liquid nitrogen and subsequently cut down by a sharp blade. The surface area and cross section structures of the papers were observed with 5 kV by using SEM (Hitachi SU8020, Tokyo, Japan) at 200× and 100× magnification, respectively. The paper samples were cut (1 × 1 cm) and mounted onto a bronze stub with carbon double-sided tape. Prior to visualization, the papers were spluttered with gold using a sputter coater (E1030 Ion Sputter, Hitachi, Tokyo, Japan).

#### 2.7.5. Water Barrier of RSPs

Water absorption of paper was investigated with a Cobb Sizing Testing (Kumagai Riki Kogyo Co. Ltd., Tokyo, Japan). Water absorption (%) was calculated from the weight of paper before and after absorbing deionized water (23 ± 1 °C) into the surface of paper within 2 min.
Water absorption (%) = [W_w_ − W_d_/W_d_] × 100(5)
where W_w_ and W_d_ are the wet and dry weights of the paper samples, respectively.

#### 2.7.6. Antibacterial Activity of Rice Straw Paper

Antibacterial activity of the RSP coated with longan extract was investigated for inhibitory effects on the test bacteria using the paper disc diffusion method. A suspension of the test bacteria (0.1 mL of 10^8^ CFU/mL) was spread on the sterile Mueller–Hinton agar plate. The RSP discs prepared by cutting to a size of 6 mm diameter were placed above the top agar. All culture plates were incubated for 24 h at 37 °C. Zones of inhibition were measured using digital Vernier caliper. The resulting inhibition zones were measured in millimeters and recorded. Three replicates were carried out for each coated or non-coated paper against each of the test bacteria. Penicillin G (10 μg) and Streptomycin (10 μg) of Oxoid™ Antimicrobial Susceptibility Discs, Thermo Fisher Scientific Inc., Waltham, MA, USA were the antibiotics used as positive test control.

### 2.8. Statistical Analysis

All experiment data of the paper samples were calculated from at least 9 replicates and expressed as mean ± SD. Analysis of variance (ANOVA) was carried out by Duncan’s multiple-range test (DMRT) using the SPSS software (SPSS for Windows, SPSS Inc., Chicago, IL, USA) at *p* ≤ 0.05.

## 3. Results and Discussion

### 3.1. Fiber Morphology and Mechanical Properties of Rice Straw Fibers

To start with, fiber morphology and mechanical properties of rice straw fibers were examined as quality parameters for paper production. The microstructure of rice straw fibers was observed through SEM image at 400× magnification (Figure 1). The open structure of rice straw allows easy diffusion and penetration of chemicals during cooking at atmospheric temperature [21]. As shown in Figure 1, fairly good rice straw fibers were obtained by cooking with diluted alkaline solution of 20% (*w*/*v*) NaOH at 100 ± 10 °C for 3 h.

As our previously reported [17] the average length and average width of rice straw fiber are summarized in Table 1. Similar morphology of rice straw fiber was also reported with an average length of 1400 μm and average fiber width was 13 µm [22]. Like other woody materials, it has been demonstrated that rice straw cell walls are also constituted of cellulose chains which are bond together with hydrogen bonds and form high tensile strength microfibrils providing further good strength properties [23].

The quality of paper is influenced by its fiber characteristics including length, diameter and lumen width [24]. It is obvious that slenderness (the ratio of fiber length to fiber width) and flexibility are dependent on the fiber attributes outlined above. The flexibility coefficient is a measure of the strength properties of paper. It gives the tensile and bursting strength of the fiber, while the slenderness ratio (the so-called Felting coefficient) is a measure of tearing strength of paper. The fibers with a high Felting coefficient are longer, thinner and have high tearing strength than fibers with a low Felting coefficient [24]. The implication of the flexibility coefficient and Felting coefficient provides more bonding area and subsequently stronger papers are produced [25]. An important factor in paper production is a reflection of the fiber cell wall thickness. The Runkel ratio of fiber was calculated from lumen width and cell wall thickness. If the Runkel ratio is more than 1, the fiber is hard and difficult to feel during the paper production, and the quality of the paper will be grossed with poor bonding. For paper making, it is suggested that the values of the Runkel ratio, flexibility coefficient and Felting coefficient should be over than 1, 0.5 and 75, respectively [26]. The results shown in Table 1 indicate that the obtained rice straw fibers can be used for paper production. Our previous research also revealed that rice straw fibers could be altered as a potential raw material for cellulose-based packaging [17].

### 3.2. HPLC Analysis and Total Phenolic Content of Longan Peel Extract

The obtained longan peel extract prepared by extraction using a mixture of ethanol and water is regarded as environmentally friendly process. Total phenolic content of 13.21 μg GAE/mg was determined in the longan peel extract. Major individual phenolic compounds of the longan peel extract included gallic acid (1.36 μg/mg), corilagin (2.43 μg/mg) and ellagic acid (3.85 μg/mg). These bioactive compounds available in longan pericarp or peel were also reported by other researchers [3,5].

### 3.3. FTIR Spectra of Longan Peel Extract Non-Coated and Coated RSP

The obtained longan peel extract was further examined using spectroscopic analysis. Infrared spectroscopy works based on the atoms’ vibrations in a molecule to be tested. When the test molecule absorbs infrared radiation, the chemical bonds in its vibrate and are able to stretch, contract or bend. Figure 2 shows the spectra of the longan peel extract, RSP and 20% (*w*/*v*) longan extract coated RSP in the wavenumber range from 400 to 4000 cm^−1^. In Figure 2A, the FTIR spectra of the crude extract at 511 cm^−1^ and 2920 cm^−1^ were assigned to C–H vibrations of alkene. Bands in the range 1500–1300 cm^−1^ are also frequently ascribed to symmetric and antisymmetric deformational vibrations of C–H in methyl (CH_3_) groups. The peak displayed the main absorption at 3305 cm^−1^, which is typically ascribed to components with O–H bond of carboxylic acid or phenol. The band at 1604 cm^−1^ is originated from C=C of alkene, whereas another band was observed at 1024 cm^−1^ for C–O stretching of ether. FTIR analysis demonstrated the functional groups of phytochemicals and provided a preliminary characterization of the peel extract. Ellagic acid, quercetin and kaempferol of flavone glycosides, and complex of hydroxycinnamates in longan peel extract were investigated and reported as phenolic compounds [27].

As shown in Figure 2B, similar observations have been reported for the alkaline pulping of rice straw [28]. The main absorption band, at approximately 3330 cm^−1^, may be due to various hydroxyl (O–H) stretching vibrations. In addition, the main peak detected at 1040 cm^−1^ was attributed to the ether linkage (C–O). The shoulder peak at 890 cm^−1^ was distinguished in rice straw paper (Figure 2B). This occurrence has been described through infrared analysis as amorphous cellulose [28]. It was observed that the range 890–1430 cm^−1^ is the typical FTIR spectrum of cellulose [16]. The coating of longan peel extract onto RSP was mainly visible by intensity increase in bands arising at 2920 cm^−1^ and 1604 cm^−1^, which are characteristics for the C–H and C=C assignment of the peel extract (Figure 2C). This observation would confirm that the longan peel extract was successfully coated onto RSP.

### 3.4. Characterization of Non-Coated and Coated RSP

#### 3.4.1. Physical Properties of Non-Coated and Coated RSP

To produce sustainable natural packaging, RSP samples were prepared without further chemical bleaching process. The pulping process used in this study was found to be better in terms of reducing the chemical contaminants as well as the pollution load of bleaching effluents. It has been reported that rice straw contains less lignin content than softwoods and hardwoods; therefore, it requires mild cooking conditions for pulp production [8]. Figure 3 illustrates the visual aspect of the prepared RSPs. It can be observed that RSP represented the residual lignin content within the pulp, which was responsible for color (Figure 3A).

The addition of the longan peel extract also influenced on appearance and brightness of the RSP. The brightness of the obtained papers relatively decreased from 34.15% to 16.05% after coating the RSP with the longan peel extract (Table 2). The coated RSP developed an intense brownish color, which was mainly due to the components of longan peel extract (Figure 3C–E). Table 2 shows that addition of 20% longan peel extract gave a significant increase in thickness and the basic weight of the coated papers was comparable to those of the control papers. It was found that coating of longan peel extract at 20% (*w*/*v*) affected the tensile index of RSP. This result might therefore be influenced by the matrix of the extract retained in the internal porous space of RSP after the coating process which resulted in an increased tensile index of the coated papers.

The obtained results of this study showed comparable tensile index to our previous report on biosynthesized poly(3-hydroxybutyrate) coated pineapple leaf fiber paper [29], soda pulping of rice straw paper [8], sunflower stalk paper and industrial made paper [30].

#### 3.4.2. Water Barrier Properties of Non-Coated and Coated RSP

The barrier of water absorption of RSP was further studied in order to investigate its ability to be used as packaging material. Coatings of longan peel extract in different concentrations on RSP led to the improvement of water resistance of the obtained coated RSP, as indicated by relatively low water absorption values (Figure 4). This effect could be attributed to the hydrophobic nature of phenolic compounds in longan peel extracts. Structurally, the alkyl and halogen groups of phenolic compounds enhance hydrophobic character of benzenoid rings, thereby decreasing water solubility [31]. This result clearly demonstrated that hydrophobic longan peel extract could delay water absorption into RSP matrix. For better water resistance, it may be necessary to modify the paper sample to achieve a denser structure. Nevertheless, the obtained coated paper can be sufficiently used for the packaging of dry foods, such as baked goods, that does not require an excellent barrier to water.

#### 3.4.3. Scanning Electron Microscopy (SEM) of RSP

SEM was carried out to visualize and confirm the changes of surface and cross-section structures of papers with and without coating. The microstructures of selected papers are shown in Figure 5.

Relatively low water absorption was confirmed by the structure of coated RSPs. Microstructures of the RSP coated with oxidized starch and coated with longan peel extract (Figure 5B,C) appeared to be homogeneous, with the mean thickness of 440 ± 60 and 380 ± 20 μm, respectively. On the other hand, it is possible to see porous structures on the surface of non-coated RSP (Figure 5A). Visible clusters of rice straw fibers were observed in uncoated RSP by cross-section SEM (Figure 5A). The surface structure virtualized by SEM also confirmed an increase in the tensile index of the coated RSP due to the extract being retained in the internal porous space of RSP.

### 3.5. Antibacterial Activity of RSP with and without Longan Peel Extract Coating

As shown in Table 3 and Figure 6, the coated RSPs with longan peel extract showed antibacterial activity against Gram-positive bacteria, *S. aureus* and *B. cereus*. However, it had no antibacterial activity on the test Gram-negative bacterium, *E. coli*. This is due to the specific antibacterial action on Gram-positive bacteria. The ability of gallic acid, ellagic acid and (−)-epicatechin on the disruption of the molecular structure of *S. aureus* cell, could be one of the possible factors that contribute to their antimicrobial activities [7]. The use of natural essential oils of clove, cinnamon and oregano coating on paper packaging materials has been reported, but they did not prevent the growth of Gram-positive bacteria (*B. cereus*, *S. aureus, Listeria monocytogenes* and *Enterococcus faecalis*,) [32]. Without coating of longan peel extract, the non-coated-RSP had no antimicrobial activity (Table 3).

The biosynthesis of silver nanoparticles using aqueous of longan peel have been reported for the activity to prevent the growth of bacteria, but there was no antibacterial activity on *Pseudomonas aeruginosa*, *E. coli*, *Bacillus subtilis* and *S. aureus* when using only the aqueous longan peel extracted by distilled water [33,34]. In contrast to peel extract, the antibacterial activity was observed in longan seed extracted by a mixture of ethanol and water. The seed extract exhibited antibacterial activity against *Propionibacterium acne*, *Streptococcus mutans*, *S. aureus*, *Acinetobacter baumannii* and *Salmonella gallinarum* [18].

The polarity of the solvent and that of the different phenolic compounds affect extraction efficiency and, therefore activity of the obtained extracts. The most polar phytochemicals can be extracted using water. Generally, no single solvent will provide optimum recovery of all phenols or even a limited range of phenolic compounds. Sample preparation and the extraction processes of phytochemicals from tropical and subtropical fruit biowastes have been extensively reviewed [2].

Comparing the longan peel extract coated RSP with the positive control or antibiotics [17], the results showed that the coated RSP had similar activity to inhibit the growth of pathogenic bacterium *B. cereus* to that observed in Penicillin G (Table 3). On the other hand, Streptomycin had a negative result on the growth of food pathogenic bacterium, *B. cereus*. Based on the size of the inhibition zone, the RSP containing longan peel extract exhibited bacterial activity higher than pomelo peel extract coated RSP [17] for more than 50% and 60% on *S. aureus* and, *B. cereus*, respectively.

Paper is generally used as packaging for dry food e.g., snacks and fried food products. As such, *B. cereus* is a microbe commonly contaminating in dry food including carbohydrate and cereal-based products [35]. Like *B. cereus*, it has been reported that *S. aureus* are generally found in poor personal hygiene of food handlers. Various types of foods serve as an optimum growth medium for *S. aureus*. Foods that have been frequently contaminated by *S. aureus* are meat and meat products, poultry and egg products, dairy products, salads, baked products (especially cream-filled pastries and cakes), and sandwich fillings [36]. Our RSP coated by longan peel extract displaying antibacterial activity against *B. cereus* and *S. aureus* can be alternatively used as paper packaging to prevent bacterial contamination of dried food products.

## 4. Conclusions

Recently the peel and seed of fruit have been of growing interest due to their biological activity, and the current study focuses on the possibility of using peel waste as a source of low-cost, natural antibacterial biocompounds. Fast growing plant species have become more promising as alternative non-wood fiber sources to solve the gap between wood fiber demand and supply. Although there are a variety of biopolymers available, cellulose still stands out as a practical selection for packaging beyond its current use in boxes and paper bags. This study revealed that rice straw residues can be used as an alternative non-wood raw material for paper production using eco-friendly processes, and especially for disposal packaging which has a relatively shorter life cycle. The longan peel extracts coated RSP exhibited greater water barrier and antibacterial activity as compared to control papers. The activity of coated RSP against food pathogenic bacteria, *Staphylococcus aureus* and *Bacillus cereus* indicated the potential uses for active packaging. The coating of bioactive compounds can be effectively applied to other cellulose-based materials.

## Figures and Tables

**Figure 1 polymers-13-03096-f001:**
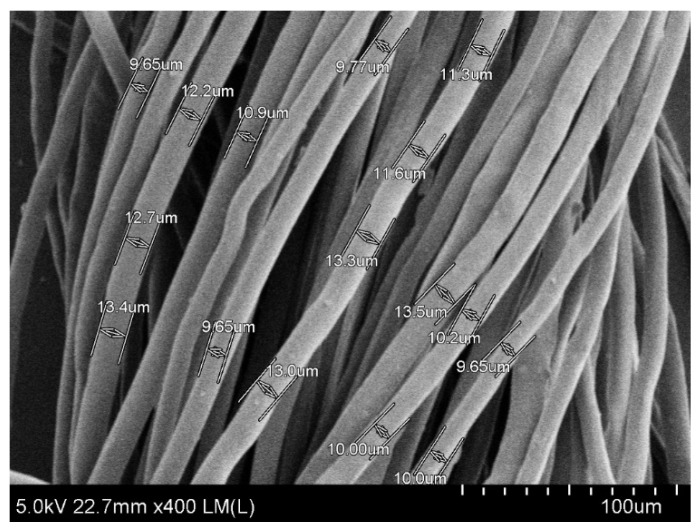
SEM images of microstructure of rice straw fibers at 400× magnification.

**Figure 2 polymers-13-03096-f002:**
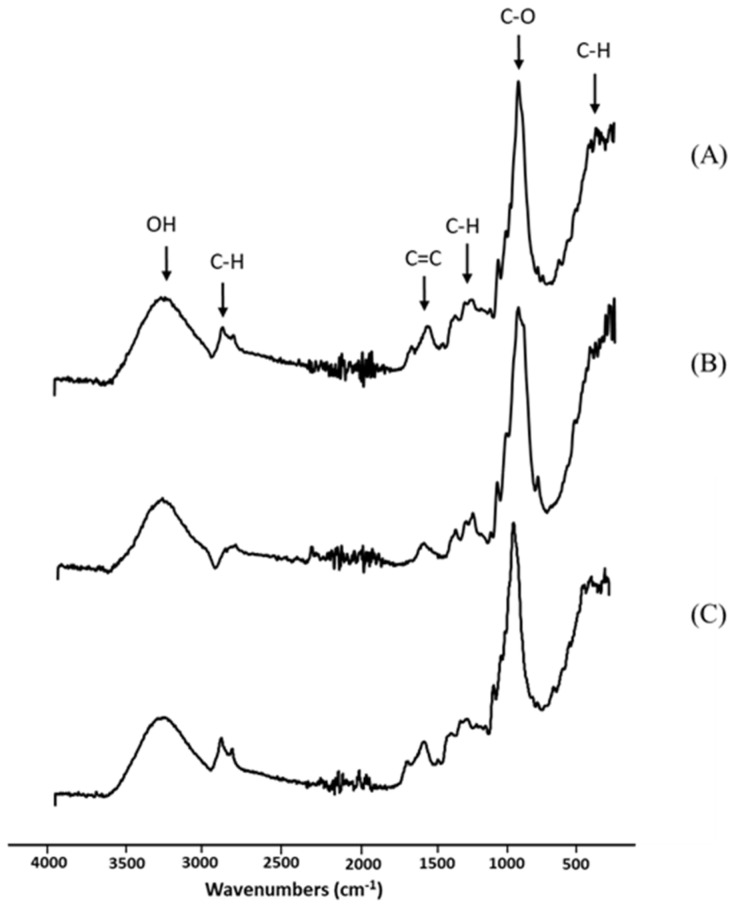
Fourier transform infrared (FTIR) spectra of longan peel extract (**A**); non coated RSP (**B**) and 20% (*w*/*v*) longan peel extract coated RSP (**C**).

**Figure 3 polymers-13-03096-f003:**
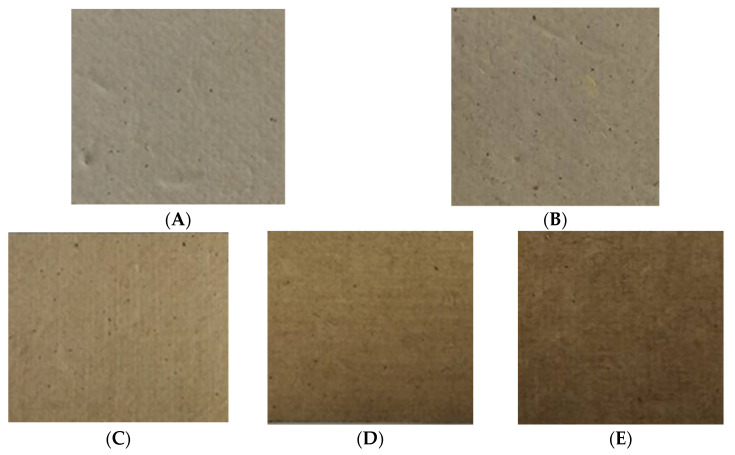
Appearance of RSP without coating (non-coated RSP) (**A**); coating with commercial hydrophobic starch (starch coated RSP) (**B**); coating with 10% (*w*/*v*) (**C**); 15% (*w*/*v*) (**D**); and 20% (*w*/*v*) of longan peel extract (coated RSP) (**E**).

**Figure 4 polymers-13-03096-f004:**
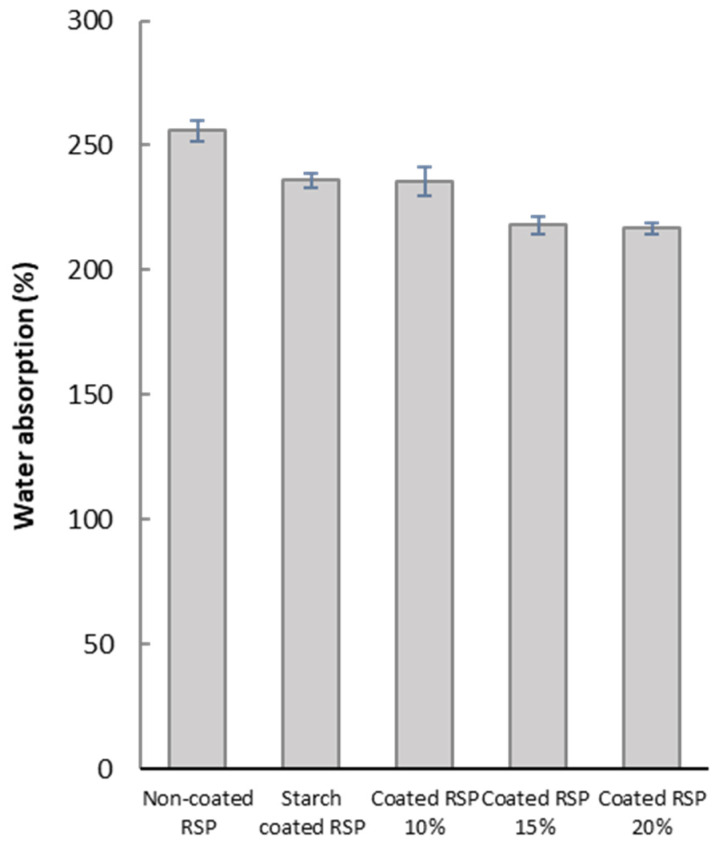
Water absorption of the non-coated RSP, starch coated RSP and RSP coating with longan peel extract at 0%, 10%, 15% and 20% (*w*/*v*).

**Figure 5 polymers-13-03096-f005:**
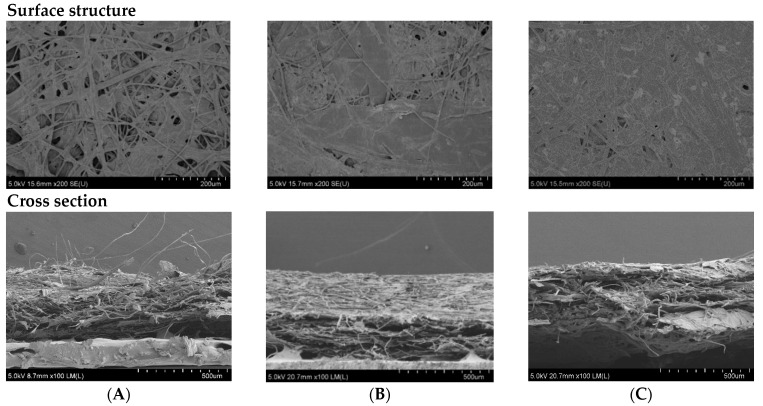
SEM micrographs of surface structure (at 200× magnification) and cross section (at 100× magnification) of the non-coated RSP (**A**); coating with commercial hydrophobic starch (starch coated RSP) (**B**) and coating with 20% (*w*/*v*) longan peel extract (coated RSP) (**C**).

**Figure 6 polymers-13-03096-f006:**
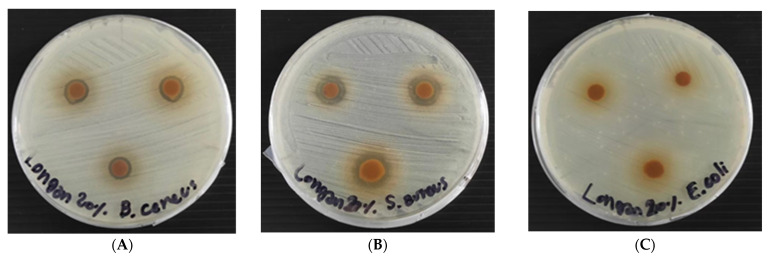
Antibacterial activity of coated RSP with 20% (*w*/*v*) longan peel extracts on *Staphylococcus aureus* (**A**); *Bacillus cereus* (**B**) and *E. coli* (**C**) using disc diffusion method.

**Table 1 polymers-13-03096-t001:** Fiber morphology and mechanical properties of rice straw fiber.

Sample	Fiber Morphology				Mechanical Properties		
	This Study	Reference [17]			This Study	Reference [17]	
	Cell Wall Thickness (μm)	Fiber Length (μm)	Fiber Width (μm)	Lumen Width (μm)	Runkel Ratio	Flexibility Coefficient	Felting Coefficient
Rice straw fiber	1.5 ± 0.1	800 ± 410	10.2 ± 5.3	8.7 ± 0.4	0.34 ± 0.01	0.85 ± 0.04	78.43 ± 4.12

**Table 2 polymers-13-03096-t002:** Properties of the non-coated RSP, starch coated RSP and the longan peel extract coated RSP at 10%, 15% and 20% (*w*/*v*).

Properties	Non-Coated RSP [17]	Starch Coated RSP	Coated RSP with Longan Peel Extract (*w*/*v*)		
			10%	15%	20%
Basis weight (g/m^2^)	106.42 ± 6.08 ^b^	118.19 ± 5.93 ^a^	118.20 ± 9.44 ^a^	117.03 ± 9.95 ^a^	118.95 ± 3.32 ^a^
Thickness (mm)	0.34 ± 0.01 ^c^	0.44 ± 0.06 ^a^	0.38 ± 0.01 ^b^	0.39 ± 0.07 ^b^	0.38 ± 0.02 ^b^
Brightness (%)	34.15 ± 0.49 ^a^	32.76 ± 1.64 ^a^	22.74 ± 1.39 ^b^	20.76 ± 0.85 ^c^	16.05 ± 0.62 ^d^
Tensile index (N·m/g)	32.26 ± 4.36 ^b^	33.05 ± 4.92 ^b^	32.98 ± 5.78 ^b^	32.67 ± 3.15 ^b^	36.20 ± 2.75 ^a^

Values with different letters in each row are significantly different (*p* < 0.05). ^a–d^ Different letters in each row are significantly different (*p* < 0.05) when using one-way ANOVA.

**Table 3 polymers-13-03096-t003:** Antibacterial activity of the non-coated RSP and coating with longan peel extract at 10%, 15% and 20% (*w*/*v*) against food pathogenic bacteria.

Bacteria Test	Diameter of Inhibition Zone (mm) ^1^					
	Non-Coated RSP	RSP with Longan Peel Extract (*w*/*v*)			Positive Control ^2^ [17]	
		10%	15%	20%	Penicillin G	Streptomycin
*S. aureus* ATCC 6538	-	12.57 ± 1.00 ^d^	14.36 ± 0.37 ^c^	14.47 ± 1.47 ^c^	43.20 ± 2.02 ^a^	18.50 ± 0.50 ^b^
*B. cereus* ATCC 11778	-	10.37 ± 1.01 ^a^	11.10 ± 0.62 ^a^	10.77 ± 0.63 ^a^	11.50 ± 0.50 ^a^	-
*E. coli* ATCC 25922	-	-	-	-	8.00 ± 0.50 ^b^	23.00 ± 0.87 ^a^

^1^ The diameter of bacteria inhibition (mm), including the diameter of the disc 6 mm. ^2^ Penicillin G of 10 µg was used as positive control for *S. aureus* and *B. cereus*. Streptomycin 10 µg was used as positive control for *E. coli*. The results of the experiments determined by the different letters in each row are significantly different (*p* < 0.05). ^a–d^ Different letters in each row are significantly different (*p* < 0.05) by using one-way ANOVA. - Means no inhibition of bacteria growth.

## Data Availability

Data are contained within the article.

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
