# Peer review of "Eco-Friendly Rice Straw Paper Coated with Longan (Dimocarpus longan) Peel Extract as Bio-Based and Antibacterial Packaging"

_polymers, 2021, doi:10.3390/polym13183096_

Round 1

Reviewer 1 Report

In this paper “Eco-Friendly Rice Straw Paper Coated with Longan (Dimocarpus longan) Peel Extract as Bio-based and Antibacterial Packaging”, the authors reported an active paper from rice straw fibers with its function as antibacterial activity obtained from longan (Dimocarpus longan) peels. The paper fit the aims and scope of Polymers. I would recommend accepting the paper after modifications.

  1. FTIR and SEM should be defined in Abstract.
  2. Introduction should be substantially improved to clarify the novelty of the manuscript. There are so many reports on sustainable and functional materials. The author should explain why it is interesting to do the experiments they describe and especially what is new compared to these published papers. In my opinion, source of raw materials which can be obtained from agricultural and industrial processing by-products and wastes, is worth to be emphasized. The reuse of waste or by-products in agro-food industry can increase economic value and environmental benefits and better highlight sustainability. It might be better to simplify and better explain with realistic examples to evidence the need to reuse agro-food waste by-product.

doi: 10.1016/j.lwt.2021.111617, doi 10.1021/acs.jafc.0c00945, doi:10.3390/foods8080286

  1. The development of bio-based and antibacterial packaging materials should be introduced. For example

doi:10.3390/foods9040449, doi:10.1016/j.jfoodeng.2021.110697, doi:10.1016/j.foodhyd.2018.11.051

  1. The method of FTIR should be described in details. Were KBr used in the method? Please confirm the range of 4000-500 cm-1
  2. it is strongly suggested to indicate the purity and amount of Penicillin G and Streptomycin in Section 2.6.
  3. How about the antibacterial activity of the uncoated RSP?

Reviewer 2 Report

Presented work should be completed in some points:

  • Introduction should be completed by the review of antibacterial agents found in different sources (extracts),
  • Chemical composition of the extract should be investigated (using chromatographic techniques). In the introduction several components are listed, so presence of some of them should be confirmed,
  • Obtained results should be compared with other materials dedicated for packaging industry.

Reviewer 3 Report

I would like to appreciate the author's effort towards developing a new active packaging blend. I find it even more relevant given that it is biodegradable and made upon discarded agri-food waste. Thus not only the developed material could be biodegradable and functional but also contribute to a circular economy model and reduced pollution. The authors tested the physical properties and potential antibacterial activity of the material on two common foodborne pathogens. Their results suggest that this material could be used for its proposed purposes. The work is generally well-written and sound, and I find its presentation correct. There are, however, some minor changes that should be made:

L34: It would be best to mention that longan is an East Asian fruit in order to provide a broader perspective to the reader.

L42-45: A brief yet concise relation between polyphenolic content and antioxidant or antimicrobial properties should be established. This is, making clear that these compounds are highly liable for these bioactive properties.

L54: Oxidation and its relation to bacterial growth in foods should at the very least be mentioned.

L59-60: It is being hinted that these molecules can be used as antibiotics, not food preservatives. One thing is to inhibit microbial growth and other very different, potential use as antibiotics. Please rephrase and make clear the proper uses these discarded matrices could have as food additives, with rigor.

Table 1: Please keep the presentation order from the previous study.

After these minor changes are made, I would consider the work ready for publication. Therefore, I am suggesting MINOR REVISIONS.

Round 2

Reviewer 2 Report

Work was improved in comparison to previous submission, so in my opinion it can be accepted in present form. Thank you very much for the provided corrections and completions.